# Drivers of land-use changes in societies with decreasing populations: A comparison of the factors affecting farmland abandonment in a food production area in Japan

Yoshiko Kobayashi[1¤]*, Motoki Higa[2], Kan Higashiyama[1], Futoshi Nakamura[1]

1 Research Faculty of Agriculture, Hokkaido University, Sapporo, Hokkaido, Japan, 2 Faculty of Science, Kochi University, Kochi, Kochi, Japan

¤ Current address: Western Region Agricultural Research Center, National Agriculture and Food Research Organization (NARO), Zentsuji, Kagawa, Japan
* mail.yoshikoba@gmail.com

**Data Availability Statement:** All relevant data are within the paper and its Supporting Information files.

## Abstract

The extraordinary population growth of the 20th century will subside in the 21st century, followed by depopulation, constituting the first population decline phase in human history in Japan and other developed countries. The drivers of land-use change during the population decline phase are expected to differ from those of the population growth phase; however, research on land-use drivers during the decline phase is limited. Identifying these drivers is necessary to develop effective management plans for biodiversity and ecosystem services in the decline phase. First, we calculated the probability of farmland abandonment in Hokkaido, a Japanese food production area, from 1973–2009 and divided the period into the population growth phase (1978–1997) and the decline phase (1997–2009). We examined various geographical and social factors that were assumed to alter the land use during these two phases. Geographical and social conditions are key factors in determining the probability of farmland abandonment, but their influences varied between the two phases. The farmlands located on geographically uncultivable sites, such as marginal, underproductive, narrow, and steep land, were abandoned during these phases; however, social conditions, such as the distance from densely inhabited districts (DIDs) and the population, exerted opposite effects during these two phases. Farmland abandonment occurred near DIDs (i.e., urban areas) during the population growth phase, whereas farmland abandonment occurred far from DIDs and sparsely populated farmlands during the decline phase. Farmland abandonment was strongly affected by government policy during the population growth phase, but the policy weakened during the decline phase, which triggered farmland abandonment throughout Hokkaido. The geographical and social drivers found in the present study may provide new insights for other developed countries experiencing depopulation problems.

**Funding:** Our study was supported by the Environmental Research and Technology Development Fund (4-1504, 4-1805) of the Ministry of the Environment of Japan.

**Competing interests:** The authors have declared that no competing interests exist.

## Introduction

The 20th century witnessed extraordinary population growth. The world population grew from 1.65 billion to 6 billion, increasing more than four-fold to reach 7.7 billion in 2019 [1–3]. However, the world's population growth rate is slowly falling, and population is projected to level off or decrease before the end of this century [3]. In particular, in at least 55 out of 235 countries or areas, including Japan, populations are predicted to decline between 2019 and 2050 [3]. The United Nations [2] has provided estimates showing that population decline has already been occurring in Germany since 2005 and in Italy and Japan since 2010.

Population dynamics, such as rural-urban migration, and economic growth are tightly linked to regional patterns of land-use and land-cover changes [4–6]. In the population growth phase, human activities to enhance food and fuel production are major drivers of land-use changes and affect regional biodiversity and ecosystem services [7–9]. The habitat loss and fragmentation associated with land development occur, causing the decline or extinction of terrestrial plants and animals [9]. In particular, agricultural activities have had a major effect on the extinction of native species that had adapted to natural environments before cultivation [10].

Farmlands were intensively developed during a period of population growth from the late 19th century to the early 20th century, and they were abandoned during the period of economic growth after the 1950s in developed countries [6, 11, 12]. Environmental, socioeconomic, and political dynamics contributed to increased farmland abandonment [6, 11, 13]. Following a decrease in rural populations due to urban migration, farmlands are expected to decrease and be abandoned [14–17]. An accelerating population concentration in urban areas is expected even during the population decline phase, which may result in further increases in the number of abandoned farmlands [18, 19].

To maintain or increase biodiversity and ecosystem services in the future, the recovery of ecosystems after farmland abandonment is important [4]. Grazing pressure and the biotic (e.g., seed dispersal, seed bank) and abiotic legacy (soil fertility and water fluxes) of cultivation may determine the recovery trajectory of plant communities, and their effects remain for hundreds of years after abandonment [5, 13, 20, 21]. Thus, abandoned farmlands may recover their natural ecosystems in the future and increase biodiversity and ecosystem services, such as water regulation, carbon sequestration, and recreation, at the regional level [12, 13, 15, 22]. However, in other cases, abandoned farmlands may not recover these natural ecosystems due to a lack of native seed and propagule resources, the high fertility of exotic species introduced by agriculture, and/or soil degradation and erosion; thus, they may have a negative impact on ecosystem services [5, 13, 14, 22]. In order to develop an appropriate management plan for abandoned farmlands, it is important to detect the drivers of farmland abandonment and to predict future patterns of landscape mosaics with farmland abandonment.

Farmland abandonment has been explained in relation to the geographical and social condition of farmlands [6, 11–17, 23, 24]. In particular, changes in the structure of industry following economic growth and associated rural-urban migration are leading causes of farmland abandonment at the regional level [5, 6, 12]. In the population growth phase, farmlands were developed from fertile lowlands to hillside slopes. Following urban and industrial sprawl, some farmlands were converted for urban land use, and farmlands located in uplands or steep sloping lands were abandoned [4, 6, 25, 26]. However, the land-use transition may differ during the regional population decline phase. To the best of our knowledge, few studies have analyzed the social and geographical drivers that regulate land-use changes during the population decline phase compared with those during the growth phase.

To clarify differences in the drivers of land-use change in the growth and decline phases, we identified and compared the geographical and social conditions affecting land-use changes between 1973 and 2009 in Hokkaido. Hokkaido is the northernmost large island (83,450 km$^2$) in Japan and has been developed into the largest agricultural land complex supplying food to the domestic consumers. However, the recent population decline has been quite rapid and evident in Japan, thus presenting a typical case of farmland abandonment and land-use change associated with depopulation.

## Methods

### Study area

The study area is Hokkaido, which is the northernmost of the four main Japanese islands (Fig 1). In Hokkaido, a single administrative unit oversees fourteen subprefectures. Hokkaido is located in the subpolar zone; the minimum, mean, and maximum monthly temperatures of Sapporo, the capital city, are -7.0, 8.9, and 26.4 ˚C respectively; the mean annual precipitation is 1106.5 mm; and the mean annual maximum snow depth is 100 cm (meteorological records from 1981 to 2010). The lowland natural vegetation of Hokkaido is cool temperate deciduous forests in the western area, subarctic evergreen forests in the eastern area, and wetlands in the riparian zones. In the present day, Hokkaido is positioned as a food production area and is home to a quarter of the modernized farmlands in Japan [27]; therefore, extensive farmlands have been developed around lowland areas. Hokkaido's farmlands include paddy fields (19%), upland fields (36%), and pasturelands (44%), but the proportions and types of farmlands differ among the subprefectures on the island, which reflects the variance of the climate and agricultural products in each subprefecture, e.g., rice cultivation in the south and livestock industry in the east. These farmlands have been developed according to Japan's political land-use planning

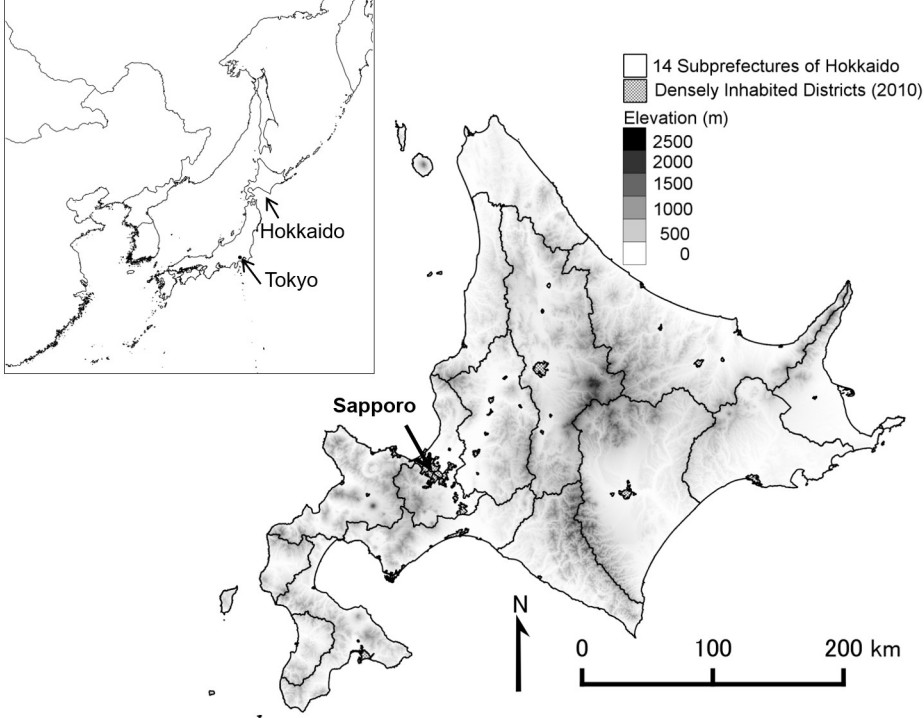

**Fig 1. Location of Hokkaido and its fourteen subprefectures.**

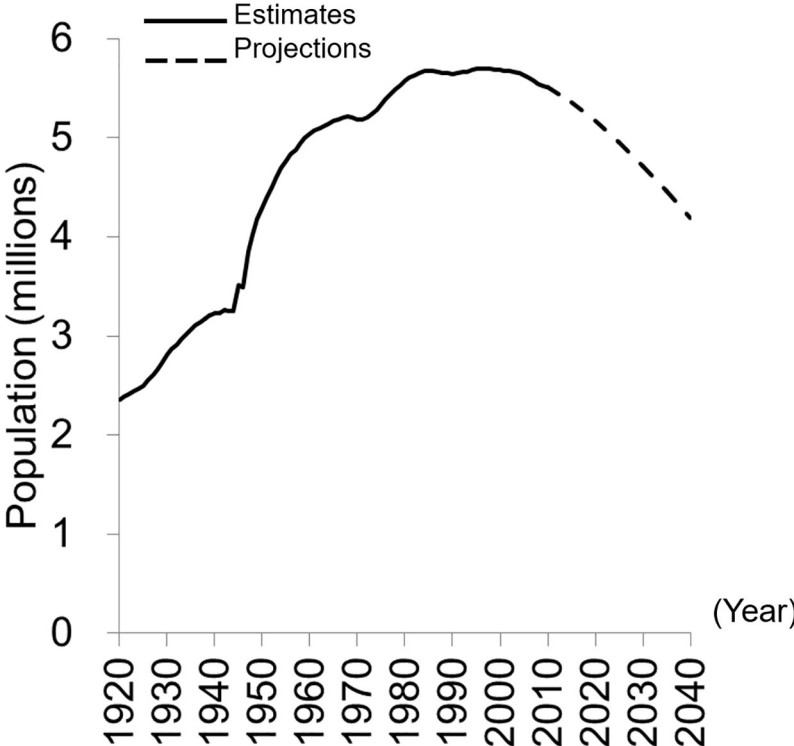

**Fig 2. Total population of Hokkaido, 1920–2010 (estimates) and 2015–2040 (projections), provided by the Statistics Bureau, MIC of Japan (2012), and National Institute of Population and Social Security Research (2013).**

since 1869. In particular, since 1952, a comprehensive development plan has been established in Hokkaido and has been renewed every 5 to 10 years, the so-called "Hokkaido Comprehensive Development Plan (HCDP)," which has promoted the expansion, modernization, and reorganization of farmlands. This plan aimed to develop agriculture, industry, and tourism in Hokkaido and reflects the history of the main industries and land use development in Hokkaido. The first term (1952–1962) focused on natural resource development such as coalmine and timber production, followed by the second term (1963–1977) emphasizing the invitation and promotion of large-scale industrial plants. The development of resorts was promoted in the 1970s, and environmental conservation was emphasized in the 2000s.

These political land-use programs have promoted resettlement to Hokkaido from the Japanese mainland and have increased Hokkaido's population. However, its population is already in decline after peaking in 1997 (Fig 2; [28, 29]). At present, depopulation in Hokkaido is progressing more rapidly than in any other region of Japan [30].

## The localization of farmland abandonment and response variables

To identify abandoned farmlands in the phases of population growth and decline in Hokkaido, we prepared digitized land-use data and land utilization segmented mesh data for 1976, 1987, 1997, 2006, and 2009 [31–35]. These land-use data were developed using topographic maps (Geospatial Information Authority of Japan) and satellite image data (Landsat, TERRA (Aster), ALOS and etc.), and they classified nationwide land usage (e.g., paddy fields, fields, orchards, forests, waste areas, building use, trunk transportation lands, lakes, and rivers) in 100-m × 100-m cells. We reclassified these data into four categories: (1) farmlands (including

all fields, orchards, and pastures); (2) forest and waste areas; (3) water areas; and (4) others (e.g., building use). Farmland transitions were identified by comparing land-use changes in the same area during each of the following four periods: (1) 1978–1987, (2) 1987–1997, (3) 1997–2006, and (4) 2006–2009. Consequently, in each period, we detected abandoned farmlands that transitioned from farmlands to forest or waste areas and developed farmlands that transitioned from forests, waste areas, or water areas to farmlands.

The farmland abandonment rates per square kilometer were calculated as a percentage of abandoned 100-m × 100-m unit cells within a 1-km x 1-km grid as follows:

$$qab_{ij} = y_{ij}/y0_{ij}$$

where $qab_{ij}$ is the ratio of abandoned farmlands in the $i$th period at location $j$; $y_{ij}$ is the number of abandoned farmlands at the end of the $i$th period at location $j$; and $y0_{ij}$ is the number of farmlands at the start of the $i$th period at location $j$.

The farmland development rates per square kilometer were calculated as follows:

$$qde_{ij} = z_{ij}/z0_{ij}$$

where $qde_{ij}$ is the ratio of developed farmlands in the $i$th period at location $j$; $z_{ij}$ is the number of developed farmlands at the end of the $i$th period at location $j$; and $z0_{ij}$ is the number of farmlands at the end of the $i$th period at location $j$.

The four periods can be divided into the growth phase and the decline phase according to the population dynamics (Fig 2); i.e., the former two periods belong to the growth phase and latter two periods belong to the decline phase. To overlay these abandonment rates on the following explanatory variables, especially the population dataset, we should consistently use the same geographical summary grids. The land-use datasets were released in 1976, 1987, 1991, 1997, 2006, 2009, 2014, and 2018, and the population data generally have been released at five-year intervals since 1920 in Japan; however, in 2010, the coordinate reference system of the geographical summary grids of population data changed to the system commonly used in the world. Therefore, we used the land-use datasets released until 2009.

## Collection of explanatory variables and statistical test

The geographical and social conditions of farmlands are key factors in determining farmland abandonment [11, 15, 16]. To determine the drivers of farmland abandonment in each period, we collected geographical and social variables of farmland conditions that are potential explanatory variables (Table 1). All the following calculations were conducted using a 1-km × 1-km grid. To obtain a median, ratio, or mode within a grid, we subdivided the grid into 100 cells (one cell is 100 m × 100 m). The data sources and collection methods were as follows:

*Area of farmlands*: the median of clumps of farmland cells within a grid.

*Ratio of central farmlands*: the proportion of central farmlands within a grid. The central farmland cell is defined as a farmland cell bordered by 4 neighboring farmland cells. The ratio was calculated as the number of central farmland cells divided by the total number of farmland cells.

*Slope*: the median of slopes of farmland cells within a grid. A cell's slope was calculated using a digital elevation model of a 10-m grid [36].

*Fertility*: the mode of ranking the productivity of farm soil [37] within a grid. The rankings ranged from 1 (good) to 4 (poor) and 5 (no data, i.e., no farm soil). An integrated criterion

**Table 1. Geographical and social variables explaining farmland abandonment in each period.**

| | Mean (±SD) | Evaluated year(s) | Source |
|---|---|---|---|
| **1976–1987 (Baseline: 1976 farmlands)** | | | |
| Area of farmlands (ha) | 7028.79 (±18449.24) | 1976* | LUM |
| Ratio of central farmlands | 0.35 (±0.28) | 1976* | LUM |
| Slope (˚) | 4.06 (±3.85) | 2009 | DEM |
| Fertility (1: good; 2: slightly negative; 3: negative; 4: poor; 5: no data (i.e., does not have cultivated fields)) | 3.28 (±1.10) | 1959–1978 | SVD |
| Ratio of roadless farmlands | 0.19 (±0.25) | 2008 | RLB |
| Ratio of past wetland farmlands | 0.04 (±0.15) | 1920s, 1950s, 2000s | MAP |
| Distance from DID (100 m) | 153.83 (±113.52) | 1980* | DID |
| Change in distance from DID (100 m) | 7.82 (±57.72) | 1980 vs. 1970* | DID |
| Population (people) | 89.10 (±427.37) | 1980* | POP |
| Population change (people) | 12.18 (±187.77) | 1980 vs. 1970* | POP |
| **1987–1997 (Baseline: 1987 farmlands)** | | | |
| Area of farmlands (ha) | 6365.39 (±12645.37) | 1987* | LUM |
| Ratio of central farmlands | 0.35 (±0.28) | 1987* | LUM |
| Slope (˚) | 4.08 (±3.77) | 2009 | DEM |
| Fertility (1: good; 2: slightly negative; 3: negative; 4: poor; 5: no data (i.e., does not have cultivated fields)) | 3.28 (±1.09) | 1959–1978 | SVD |
| Ratio of roadless farmlands | 0.20 (±0.25) | 2008 | RLB |
| Ratio of past wetland farmlands | 0.04 (±0.15) | 1920s, 1950s, 2000s | MAP |
| Distance from DID (100 m) | 161.79 (±118.33) | 1990* | DID |
| Change in distance from DID (100 m) | 5.39 (±34.91) | 1990 vs. 1980* | DID |
| Population (people) | 83.03 (±431.36) | 1990* | POP |
| Population change (people) | 3.77 (±104.62) | 1990 vs. 1980* | POP |
| **1997–2006 (Baseline: 1997 farmlands)** | | | |
| Area of farmlands (ha) | 7552.41 (±17109.87) | 1997* | LUM |
| Ratio of central farmlands | 0.36 (±0.28) | 1997* | LUM |
| Slope (˚) | 4.49 (±4.06) | 2009 | DEM |
| Fertility (1: good; 2: slightly negative; 3: negative; 4: poor; 5: no data (i.e., does not have cultivated fields)) | 3.36 (±1.12) | 1959–1978 | SVD |
| Ratio of roadless farmlands | 0.23 (±0.26) | 2008 | RLB |
| Ratio of past wetland farmlands | 0.04 (±0.16) | 1920s, 1950s, 2000s | MAP |
| Distance from DID (100 m) | 193.44 (±144.66) | 2000* | DID |
| Change in distance from DID (100 m) | 29.48 (±83.28) | 2000 vs. 1990* | DID |
| Population (people) | 72.68 (±395.95) | 2000* | POP |
| Population change (people) | 2.40 (±70.00) | 2000 vs. 1990* | POP |
| **2006–2009 (Baseline: 2006 farmlands)** | | | |
| Area of farmlands (ha) | 9602.32 (±19938.11) | 2006* | LUM |
| Ratio of central farmlands | 0.37 (±0.28). | 2006* | LUM |
| Slope (˚) | 4.12 (±3.67) | 2009 | DEM |
| Fertility (1: good; 2: slightly negative; 3: negative; 4: poor; 5: no data (i.e., does not have cultivated fields)) | 3.30 (±1.10) | 1959–1978 | SVD |
| Ratio of roadless farmlands | 0.22 (±0.25) | 2008 | RLB |
| Ratio of past wetland farmlands | 0.04 (±0.16) | 1920s, 1950s, 2000s | MAP |
| Distance from DID (100 m) | 198.81 (±146.86) | 2005* | DID |
| Change in distance from DID (100 m) | 6.22 (±36.50) | 2005 vs. 2000* | DID |
| Population (people) | 64.10 (±345.52) | 2005* | POP |
| Population change (people) | -0.63 (±25.00) | 2005 vs. 2000* | POP |

LUM: land utilization segmented mesh data provided by the National Land Information Division, MLIT of Japan.

DEM: digital elevation model with 10-m cell size provided by the Geospatial Information Authority of Japan.

SVD: digitized vector geometry data of 1:50,000 soil map provided by the Japan Soil Association.

RLB: digitized vector geometry data of road line provided by the Geospatial Information Authority of Japan.

MAP: historical topographic maps (1:50,000 or 1:25,000) and aerial photographs provided by the Geospatial Information Authority of Japan

DID: digitized vector geometry data of densely inhabited districts provided by the National Land Information Division.

POP: digitized mesh map of population data provided by Statistics Bureau, MIC of Japan.

*: These variables had been used for different estimated years that related to each period.

is evaluated based on, among other things, productivity, depth, and wetness. The digitized vector geometry map of fertility was converted into a 100-m × 100-m raster image and was superimposed.

*Ratio of roadless farmlands*: the proportion of roadless farmlands within a grid. The roadless farmland cell is defined as a farmland cell without roads across 8 neighboring farmland cells. The ratio was calculated as the number of roadless farmland cells divided by the total number of farmland cells. The digitized road vector data [38] were converted into the 100-m × 100-m grid raster image and were superimposed.

*Ratio of farmlands that used to be wetlands*: the proportion of farmlands that used to be wetlands of all the farmlands within a grid. The past wetlands were identified from topographic maps published in the 1920s, 1950s, and 2000s (Geospatial Information Authority of Japan) and from aerial photographs taken in the 1990s and 2000s. These maps were provided by the Laboratory of Conservation GIS, Rakuno Gakuen University, and the Hokkaido Research Organization [39].

*Distance from a densely inhabited district (DID)*: the median of the distance from a DID within a grid. The DID maps for 1980, 1990, 2000, and 2005 were created by the National Land Information Division [40–43] and were assumed to reflect farmland abandonment from 1976 to 1987, from 1987 to 1997, from 1997 to 2006, and from 2006 to 2009, considering the time lag between the change of drivers (DIDs) and farmland abandonment. The digitized vector geometry map of DIDs was converted into the 100-m × 100-m cell raster image.

*Change in the distance from a DID*: the median of the change in the distance from a DID within a grid during the following periods: from 1970 to 1980, from 1980 to 1990, from 1990 to 2000, and from 2000 to 2005 [40–44].

*Population*: the median of the potential population of farmers within a grid. Considering the commuting distance from residences to farmlands, the potential population of farmers was estimated as the population within a 500-m radius of a farmland cell. This 500 m also corresponds with the unit size of the agricultural land that was assigned to six families when they resettled there [45]. Therefore, farmers are highly likely to stay within a 500-m radius of farmland. The grid-scale population dataset was obtained from the Statistics Bureau of the MIC of Japan [31–46], and it was prorated into a 100-m × 100-m cell. The population datasets for 1980, 1990, 2000, and 2005 were assumed to influence farmland abandonment during 1976–1987, 1987–1997, 1997–2006, and 2006–2009, considering the time lag between the population change and farmland abandonment.

*Population change*: the median of potential changes in the farmer population within a grid between the following periods: 1980–1970, 1990–1980, 2000–2005, and 2005–2000 [46–50].

We assessed collinearity by calculating Pearson correlation coefficients ($r$) for each variable pair. Because all $r$ were less than 0.7, we decided to use these variables for explanatory variables as candidates for land-use drivers. The variables for each period were standardized to develop the statistical models.

All GIS operations and statistical analyses were performed using GRASS GIS 6.4.4 [51], QGIS 2.8 [52], and R.3.1.2 [53].

## Statistical models

To detect the drivers of farmland abandonment for each period, we developed a hierarchical logistic regression model and estimated regression coefficients and credible intervals as

follows:

$$y_{ij} \sim \text{Binomial}(p_{ij}, y0_{ij})$$

$$\text{logit}\,(p_{ij}) = \beta int_i + \beta cov_i \times X_{ij} + \varepsilon r_{ik} + \varepsilon s_{ij}$$

where $y_{ij}$ is the number of abandoned farmlands at the end of the $i$th period at location $j$; $p_{ij}$ is the probability of farmlands being abandoned in the $i$th period at location $j$; and $y0_{ij}$ is the number of farmlands at the beginning of the $i$th period at location $j$. $X_{ij}$ is a vector of the values of site covariates in the $i$th period at location $j$; $\beta int_i$ is the common intercept of the $i$th period; and $\beta cov_i$ is the vector of the parameter of regression coefficients. In Hokkaido, the proportions of farmland vary by subprefecture; thus, we assumed the following random intercepts: $\varepsilon r_{ik}$ is the random intercept of the $i$th period at subprefecture $k$, and $\varepsilon s_{ij}$ is the random intercept of the $i$th period at location $j$.

The parameters were estimated using the hierarchical Bayesian modeling framework and Hamiltonian Monte Carlo (HMC) sampling [54–56], a form of Markov chain Monte Carlo (MCMC) techniques [57]. The prior distributions of intercept $\beta int_i$ and fixed effects $\beta cov_i$ were assumed to follow a zero-mean normal distribution with common variances throughout the entire period. Similarly, the prior distributions of random effects $\varepsilon r_{ik}$ and $\varepsilon s_{ij}$ were assumed to follow a zero-mean normal distribution with common variances within a period. These common variances were assumed to follow a non-informative uniform distribution with mean 0 and variance 10,000. The posterior distributions of all parameters were obtained by three chain runs of 10,000 simulations after a burn-in of 1,000 samples, and they were thinned by 50 intervals using RStan version 2.5.0 [58] and R.3.1.2 [53]. The model was considered to have converged if the $\hat{R}$ values [59] of all the parameters were less than 1.1 [60].

Finally, we evaluated the posterior distribution means and 95% credible intervals of the fixed effect coefficients to detect the significant predictors for farmland abandonment.

## Results

### Frequency and distribution of farmland abandonment in the four periods

The frequency and distribution of abandoned and developed farmlands differed among the four periods (Figs 3 and 4). During the population growth phase, represented by 1978–1987 and 1987–1997, the developed farmland areas were greater than the abandoned farmland areas. By contrast, during the periods of 1997–2006 and 2006–2009, which were characterized by population decline, the relationship was the opposite; i.e., the abandoned farmland areas were greater than the developed farmland areas (Fig 3). In addition, the geographical distribution of developed and abandoned farmlands differed between the two phases. The farmlands have largely been developed with aggregated patterns in the eastern and northern regions of Hokkaido during the population growth phase, whereas no such trends were found during the population decline phase (Fig 4). By contrast, the farmlands were abandoned in 1976–1987 in the marginal areas of the developed farmlands with aggregated patterns; however, the abandoned farmlands were distributed widely over the entire Hokkaido area during the population decline phase, and abandoned farmlands were more common than developed farmlands (Fig 4). However, in the population growth phase from 1987 to 1997, farmland abandonment was limited.

### Determinants of farmland abandonment

The geographical conditions affecting farmland abandonment did not vary between the phases of population growth and decline, but social factors varied significantly (Table 2).

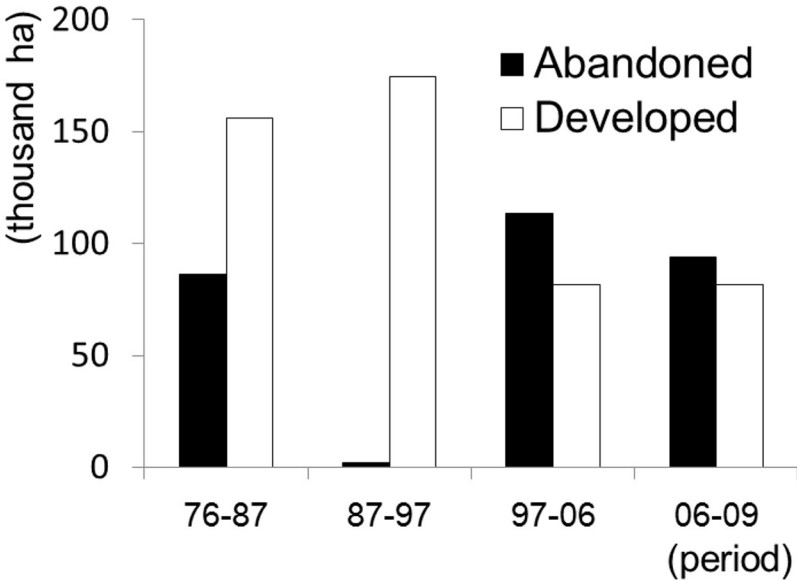

**Fig 3. Total area of abandoned or developed farmlands in the four periods.**

In the population growth phase, geographical conditions such as the ratio of central farmlands, the fertility of farmlands, and the ratio of past wetland farmlands contributed significantly to explain farmland abandonment. The slope of farmlands also explained the occurrence of farmland abandonment from 1976 to 1987 and from 1987 to 1997, but it exerted

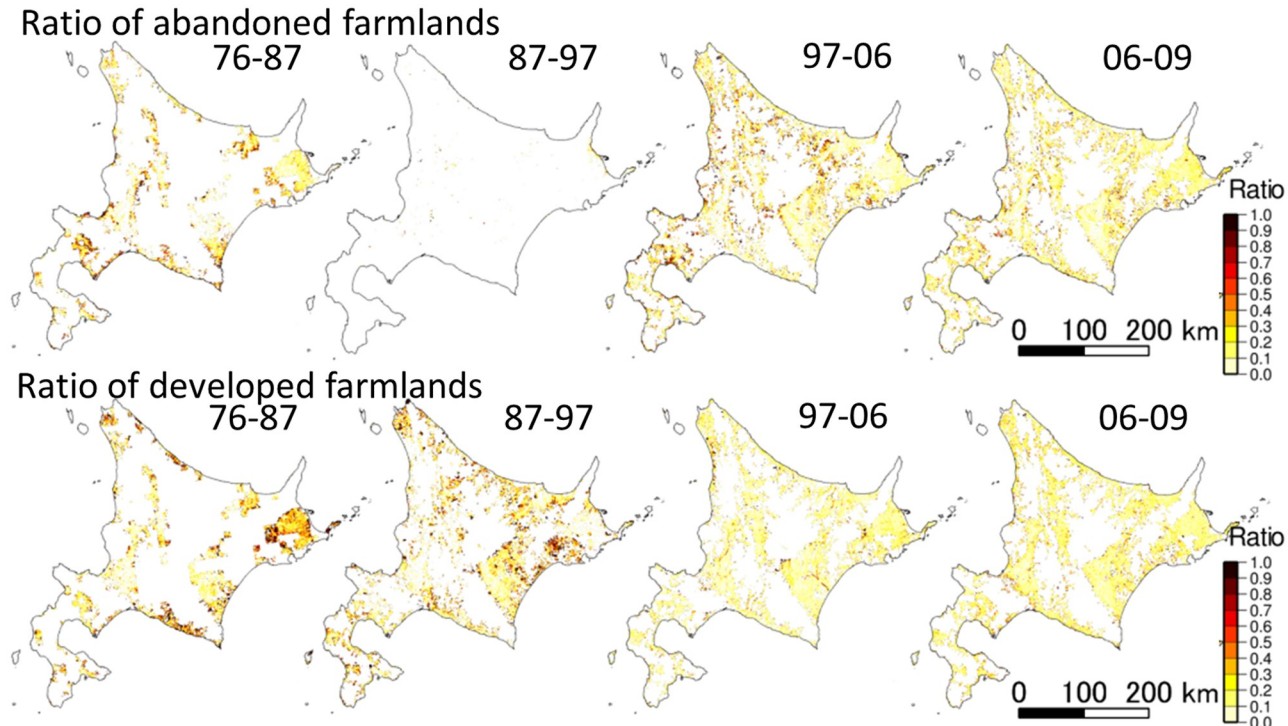

**Fig 4. Distribution maps depicting the ratio of abandoned farmlands (top) and developed farmlands (bottom) from 1976 to 2009.**

**Table 2. Posterior distribution means and 95% credible intervals for the model intercept and the coefficients of predictors for each period.**

| | Growth phase | | | | Decline phase | | | |
| --- | --- | --- | --- | --- | --- | --- | --- | --- |
| | 1976–1987 | | 1987–1997 | | 1997–2006 | | 2006–2009 | |
| Common intercept | **-4.117** | (-4.719 – -2.791) | -0.841 | (-3.071 – 1.074) | **-2.928** | (-3.187 – -2.682) | **-2.672** | (-2.799 – -2.541) |
| <Geographical variables> | | | | | | | | |
| Area of farmlands (median) | -0.024 | (-0.072 – 0.028) | **-0.286** | (-0.569 – -0.042) | **-0.148** | (-0.175 – -0.121) | **-0.025** | (-0.040 – -0.011) |
| Ratio of central farmlands | **-0.317** | (-0.371 – -0.265) | **-1.566** | (-1.814 – -1.309) | **-0.977** | (-1.004 – -0.948) | **-0.944** | (-0.963 – -0.924) |
| Slope (median) | **0.614** | (0.569 – 0.668) | **-0.502** | (-0.792 – -0.215) | **0.607** | (0.582 – 0.633) | **0.394** | (0.376 – 0.412) |
| Fertility (mode) | **0.210** | (0.164 – 0.254) | **0.350** | (0.129 – 0.566) | **0.139** | (0.116 – 0.160) | **0.057** | (0.043 – 0.071) |
| Ratio of roadless farmlands | **0.252** | (0.208 – 0.301) | 0.023 | (-0.167 – 0.204) | **0.038** | (0.014 – 0.062) | 0.015 | (-0.002 – 0.032) |
| Ratio of past wetland farmlands | **0.106** | (0.063 – 0.147) | **0.286** | (0.119 – 0.442) | **0.036** | (0.015 – 0.058) | **0.042** | (0.029 – 0.056) |
| <Social variables> | | | | | | | | |
| Distance from DID (median) | **-0.228** | (-0.278 – -0.179) | **-0.255** | (-0.501 – -0.022) | **0.248** | (0.219 – 0.275) | **0.041** | (0.026 – 0.057) |
| Change in distance from DID (median) | -0.014 | (-0.057 – 0.031) | -0.125 | (-0.453 – 0.046) | **-0.086** | (-0.111 – -0.059) | -0.013 | (-0.027 – 0.000) |
| Population within a 500 m radius (median) | 0.010 | (-0.043 – 0.063) | -0.112 | (-0.316 – 0.066) | **-0.262** | (-0.303 – -0.220) | **-0.182** | (-0.207 – -0.156) |
| Population change (median) | -0.004 | (-0.046 – 0.037) | -0.002 | (-0.085 – 0.077) | **0.029** | (0.001 – 0.057) | **-0.027** | (-0.046 – -0.009) |

Bold indicates that the posterior distribution of the coefficients did not contain zero in the 95% credible intervals.

the opposite effect in these two periods. The ratio of roadless farmlands and the area of farmlands were significant parameters from 1976 to 1987 and from 1987 to 1997, respectively. The distance from a DID was the only social condition that explained the abandonment of farmlands during the population growth phase.

In the population decline phase, geographical conditions such as the area of farmlands, the ratio of central farmlands, the slope, the fertility of farmlands, and the ratio of past wetland farmlands contributed significantly to farmland abandonment as in the growth phase. The ratio of roadless farmlands also contributed significantly from 1997 to 2006, similar to the population growth phase from 1976 to 1987. On the other hand, the influence of social conditions such as the distance from a DID, the population, and the population change significantly explained farmland abandonment, which was different from the growth phase. The distance from a DID was also a determinant of farmland abandonment, but its effect was positive in the decline phase. The population affected farmland abandonment only in the decline phase. Similarly, the population change influenced farmland abandonment only in the decline phase, but it had the opposite effect from 1997 to 2006 and from 2006 to 2009. The change in the distance from a DID explained farmland abandonment only from 1997 to 2006.

## Discussion

We can understand land-use changes and their determinants over the past 30 years when Hokkaido experienced a drastic population change from growth to decline. Our results showed that geographical and social conditions were both key factors in determining the probability of farmland abandonment, but their influences differed between the population growth phase and the population decline phase (Table 2). Geographical conditions, such as the ratio of central farmlands, have similar effects on farmland abandonment during the two phases, whereas social conditions, such as the distance from a DID and the population, have different effects in each phase. Thus, we separately discuss farmland abandonment and its drivers in Hokkaido during the population growth and decline phases.

## The determinants of farmland abandonment in the population growth phase

The frequency of farmland abandonment differed between 1976 and 1987 and between 1987 and 1997 (Fig 3), although both periods were characterized by population growth. From 1976 to 1987, abandoned farmlands frequently occurred in the marginal areas of developed farmlands. We interpreted this phenomenon to be driven by government policy, the so-called "Hokkaido Comprehensive Development Plan (HCDP)" from 1978 to 1987 [61], which promoted the relocation, concentration, and intensification of farmlands to increase agricultural productivity and farmer incomes. Consequently, infertile and inaccessible farmlands were abandoned, and aggregated, high-quality farmlands were further developed. In particular, many large-scale pilot projects were launched in the eastern and northern regions of Hokkaido, and a number of large-scale farmlands were thus developed from 1976 to 1987 (Fig 4). We should also consider the replacement of farmlands by larch plantations, whose timbers were used for the walls of the coalmine tunnels, to be a possible cause of farmland abandonment during this period [62].

By contrast, farmland abandonment was limited from 1987 to 1997, and farmlands continued to be developed at a level similar to the previous period (Figs 3 and 4). A decrease in conversion of farmlands into larch plantations due to the rapid decline of the coal industry may explain this finding [63]. We also attributed this result to the HCDP from 1988–1997 [64]. The agricultural policy in this period emphasized the rationalization and reorganization of farmlands. Small-scale farmlands were purchased and converted into large-scale modern farmlands to increase production efficiency through mechanization. Thus, both the conservation of existing farmlands and new development coexisted in this period, and the farmlands were thus rarely abandoned. In the population growth phase, farmland abandonment was tightly linked to the agricultural and forestry policy of the Japanese government.

Our study identified geographical and social determinants that promoted farmland abandonment associated with land readjustment policies. Geographical conditions, such as marginal, underproductive, narrow, steep slopes and poor road infrastructure, were determinants of increased farmland abandonment during the population growth phase (Table 2). These geographical conditions were unsuitable for the large-scale, modern farming that was encouraged by government policy. The concentration of farmland abandonment on underproductive, steep slopes and/or due to poor road infrastructure has also been observed in other countries [24–26]; in addition, government land-use policy plays an important role in shaping land-use change [6, 65–67]. From 1987–1997, a small number of farmlands located on relatively gentle slopes were abandoned, most likely because they were small and far away from an aggregated patch of large-scale farmlands. However, the limited farmland abandonment during this period makes interpreting this phenomenon difficult (Fig 4).

The distance from a DID, one of the social conditions, was an important determinant and showed a negative effect on the expansion of abandoned farmlands; i.e., the farmlands neighboring DIDs tended to be abandoned (Table 2). During the population growth phase in Japan, cities and urban environments grew outside of DIDs, and residential, commercial areas were built around DIDs because of the easy and convenient access to urban areas. Unfortunately, in Japan and other countries, this type of urbanization (urban sprawl) has been accompanied by disordered land development and land speculation. In this situation, some farmlands were developed, but they were abandoned without properly substituting other land uses. For example, Saizen et al. [66] found that farmlands have been converted into vacant spaces or forests due to urbanization from the land transition patterns in Japan's Osaka metropolitan region. Berry [68] found a similar phenomenon around the Mid-Atlantic states of New York, New

Jersey, and Pennsylvania, where farmlands around urban areas were converted into suburbs or idle lands and woodlands through land speculation.

During the population growth phase, the population was not a significant driver of the regulation of farmland abandonment in Hokkaido. A previous study indicated that industrialization in urban areas during the growth phase motivated the rural population's out-migration to urban areas, which resulted in an increase in farmland abandonment [4, 25, 26]. On the other hand, the population increase in urban areas replaced surrounding farmlands with residential areas and abandoned lands associated with urban sprawl, as indicated above [69]. The farmland abandonment driven by rural-urban migration (a decrease in the rural population) and the land abandonment driven by urban sprawl and land speculation (an increase in the suburban population) may obscure the effects of the population during the population growth phase in Hokkaido. In other words, the population might have a significant effect on land abandonment even in the growth phase, but because of the opposite (non-linear) effects of rural and suburban areas, this parameter could not be detected as a significant driver.

### The determinants of farmland abandonment in the population decline phase

In the population decline phase, farmland abandonment generally occurred more frequently than farmland development (Fig 3), and an aggregated pattern detected in the population growth phase could not be found in the decline phase (Fig 4). Large-scale farmland development and abandonment due to Japanese government policy were thus no longer in operation in this phase. The HCDP for 1998 to 2007 [70] focused on depopulation, the aging farmer population, and the increase in abandoned farmlands. The government changed its policy to use a limited budget efficiently; hence, it prioritized budget allocation for districts in which agricultural productivity could be enhanced after upgrades to irrigation and other agricultural systems. Agricultural policy thus shifted from developing new farmlands to conserving existing high-quality farmlands, which was reflected in trends of farmland abandonment and development in our study. In other words, farmlands were spontaneously abandoned with the depopulation and aging of farmers, meaning that no systematic, aggregated patterns were found in this phase.

Our results showed that when the population declined, geographical conditions, such as marginal, underproductive, narrow, steep slopes and poor road infrastructure, were also determinants of the increase in abandoned farmlands, which was a similar trend in the growth phase (Table 2). Thus, accessibility, land fertility, and landforms are important determinants of farmland abandonment, even in the decline phase, a finding similar to those of previous studies conducted in the growth phase [24–26]. Therefore, these disadvantaged farmlands will be continuously abandoned in the decline phase. The direct payment (DP) policy of subsidies has been implemented to maintain unfavorable farmlands through community-based activities since 2000 in Japan and will be effective in the decline phase. According to Shin and Kim [71], the DP implementation reduced the rate of abandonment of agricultural land by 2% in Hokkaido. In other countries as well, subsidies have been shown to be effective in reducing farmland abandonment [23, 72]. However, the current DP system has little effect on reduction of farmland abandonment if the community has lost its integrity and vitality [73, 74]. It will be necessary to determine the drivers of abandonment drivers and update the DP system continuously to maintain farmlands with a high risk of abandonment.

On the other hand, social conditions, such as the distance from a DID and the population, were other important drivers of farmland abandonment during the decline phase. The distance from a DID was positively correlated with land abandonment; i.e., the farmlands far

from DIDs tended to be abandoned. This effect was the opposite of that during the population growth phase (Table 2). The areas located far from DIDs had difficulties in accessing and delivering agricultural commodities to cities and urban areas. This low accessibility to markets will increase the transport costs of agricultural products and/or reduce the opportunities to use lands for tourist farms, which discourages farmers from continuously cultivating their lands [12, 14]. The population was identified as a significant driver only in the decline phase. We considered the labor force shortage associated with depopulation to be one of the main reasons for land abandonment in this phase [75]. In general, depopulation progresses as farmer's age. Thus, the small-scale farmlands in sparsely populated areas tend to be cultivated by elderly couples without any successors and thus cannot be maintained after farmers reach a certain age.

Population changes and the distance from a DID influence farmland abandonment differently between 1997 and 2006 and between 2006 and 2009 (Table 2). From 1997 to 2006, the preliminary stage of depopulation in Hokkaido, areas located closer to DIDs and/or areas with growing populations were likely abandoned. This trend, however, was not detected from 2006 to 2009, and areas with declining populations were likely abandoned. The effects of land-use factors, such as urbanization, which were the main drivers of farmland abandonment in the population growth phase, remained during the earliest stage of depopulation. Thus, a complex and time-lagged response of land-use and population factors (e.g., urban sprawl and land speculation) was found in this period.

Our results showed that population decline has become one of the determinants for farmland abandonment in recent years; therefore, we expect that abandoned farmlands will increase in the future with depopulation. Ohashi et al. [18] predicted a future decrease in farmlands and an increase in forests and wastelands using future scenarios of population distribution. They expected that the increase or a decrease in farmlands varies with the centralization or decentralization of the population, with decentralization mitigating farmland abandonment by approximately 10%. Thus, abandoned farmlands can be reduced by using land management policies even during the population decline phase. The land management policy for population decentralization has been initiated in the latest HCDP for 2016 to 2025 [76]. The latest HCDP would be a key policy for mitigating farmland abandonment.

Previous studies focused on the relationship between land abandonment and population or population changes and were conducted only during the growth phase without considering time-series variation. Thus, the geographical and social drivers found in the present study may provide new insights for other developed countries experiencing depopulation problems.

## Concluding remarks

We found that the determinants of farmland abandonment varied between the phases of population growth and decline. In particular, social conditions such as the population and the distance from a DID have very different effects on farmland abandonment between the two phases. The land abandonment driven by these determinants can be expected in other developed countries that may face depopulation in the near future.

While subsidies for reduction of farmland abandonment and decentralization policies may mitigate farmland abandonment, natural recovery of vegetation in abandoned farmlands will proceed in the future due to the limited financial resources and human labor to sustain unfavorable farmlands [15]. The vegetation recovery after farmland abandonment may have negative or positive impacts on biodiversity depending on conservation targets and regional circumstances [22, 77]; for example, this process may provide an opportunity to restore a quasi-natural ecosystem, such as secondary forests and grasslands [12, 13], or it may result in a

loss of specific habitats that have been sustained by traditional agricultural activities, such as paddy field wetlands, semi-natural grasslands and coppice forests [78, 79]. Furthermore, abandoned farmland is recognized as a potential threat to biodiversity by creating a "novel" ecosystem; that is, it may be converted into a completely different ecosystem dominated by specific species or into a wasteland dominated by exotic species [5, 13, 80, 81]. To prevent abandoned farmlands from transitioning into "novel" ecosystems by natural succession, we have to build conservation plans and implement appropriate restoration measures. It may be difficult to restore forest ecosystems to a composition resembling that of natural forests if there are no seed sources in the surrounding areas [82, 83]. However, open-land vegetation (e.g., grassland and wetland) can be restored due to the high seed dispersal ability of grass species [84–88].

In Japan, there are very few attempts to rewild abandoned farmlands because vegetation recovery in abandoned farmlands is considered to be a threat to sustaining agrobiodiversity represented by traditional agricultural activities [78, 79]. However, there are several activities intended to restore abandoned farmlands to original natural ecosystems in areas with high potential and needs for rewilding. On Sado island, abandoned paddies function as foraging habitats of the reintroduced Crested Ibis, *Nipponia nippon* [89], and in the Lake Inbanuma watershed, abandoned paddies on small valley bottoms have potential for multipurpose usages such as flood control, water purification, and wetland biodiversity [90]. In Hokkaido, the abandoned farmlands could be used to harbor organisms of wetland ecosystems [84–87] and provide wetland green infrastructure [91]. Natural recovery of vegetation in abandoned farmlands will proceed in the future, although the government would prefer to maintain the present farmlands through subsidies and land management policies. Thus, it is important to develop comprehensive restoration plan using natural and artificial measures to estimate the local potential to restore ecosystems. As indicated in the present study, we are carefully monitoring and predicting future land abandonment, planning multiple scenarios to project future landscape development with depopulation, and estimating the associated changes in biodiversity and ecosystem services.

## Supporting information

**S1 Data. Data from abandoned farmlands and explanatory variables for statistical models.** Column names are based on the text and Table 1. Period IDs are as follows: 1: 1976–1987; 2: 1987–1997; 3: 1997–2006; 4: 2006–2009.
(CSV)

**S1 Fig. The GeoTIFF file to position the location ID in the correct location and geometry.** Positions of location IDs are referenced in latitude and longitude based on the Tokyo geodetic reference system (EPSG: 4301).
(TIF)

## Acknowledgments

We are grateful to Prof. M. Kaneko, Dr. T. Suzuki, and Dr. S. Ono for providing digitized geometry historical wetland data.

## Author Contributions

**Conceptualization:** Yoshiko Kobayashi, Motoki Higa, Futoshi Nakamura.

**Data curation:** Yoshiko Kobayashi, Motoki Higa.

**Formal analysis:** Yoshiko Kobayashi.

**Funding acquisition:** Futoshi Nakamura.

**Investigation:** Yoshiko Kobayashi.

**Methodology:** Yoshiko Kobayashi, Motoki Higa.

**Project administration:** Futoshi Nakamura.

**Software:** Yoshiko Kobayashi, Motoki Higa.

**Supervision:** Futoshi Nakamura.

**Visualization:** Yoshiko Kobayashi, Motoki Higa.

**Writing – original draft:** Yoshiko Kobayashi, Futoshi Nakamura.

**Writing – review & editing:** Yoshiko Kobayashi, Motoki Higa, Kan Higashiyama, Futoshi Nakamura.

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
