## [Decision Letter · Decision Letter 0]

19 Mar 2020

PONE-D-20-05452

Drivers of land-use changes in societies with decreasing populations: A comparison of the factors affecting farmland abandonment in a food production area in Japan

PLOS ONE

Dear Dr Kobayashi,

Thank you for submitting your manuscript to PLOS ONE. After careful consideration, we feel that it has merit but does not fully meet PLOS ONE’s publication criteria as it currently stands. Therefore, we invite you to submit a revised version of the manuscript that addresses the points raised during the review process.

Specifically, the reviewers have pointed to the literature review and discussion sections to be strengthened and contributions to the existing research to be more explicitly discussed.

We would appreciate receiving your revised manuscript by 18 April 2020. To enhance the reproducibility of your results, we recommend that if applicable you deposit your laboratory protocols in protocols.io, where a protocol can be assigned its own identifier (DOI) such that it can be cited independently in the future. For instructions see: http://journals.plos.org/plosone/s/submission-guidelines#loc-laboratory-protocols

We look forward to receiving your revised manuscript.

Kind regards,

Eda Ustaoglu, PhD

Academic Editor

PLOS ONE

Journal Requirements:

2.  We note that Figure 2 and 5 in your submission contain map images which may be copyrighted. All PLOS content is published under the Creative Commons Attribution License (CC BY 4.0), which means that the manuscript, images, and Supporting Information files will be freely available online, and any third party is permitted to access, download, copy, distribute, and use these materials in any way, even commercially, with proper attribution. For these reasons, we cannot publish previously copyrighted maps or satellite images created using proprietary data, such as Google software (Google Maps, Street View, and Earth). For more information, see our copyright guidelines: http://journals.plos.org/plosone/s/licenses-and-copyright.

a) You may seek permission from the original copyright holder of Figure(s) [#] to publish the content specifically under the CC BY 4.0 license. 

Reviewers' comments:

Reviewer's Responses to Questions

**Comments to the Author**

1. Is the manuscript technically sound, and do the data support the conclusions?

Reviewer #1: Yes

Reviewer #2: Partly

Reviewer #3: Yes

2. Has the statistical analysis been performed appropriately and rigorously? 

Reviewer #1: Yes

Reviewer #2: Yes

Reviewer #3: Yes

3. Have the authors made all data underlying the findings in their manuscript fully available?

Reviewer #1: Yes

Reviewer #2: Yes

Reviewer #3: Yes

4. Is the manuscript presented in an intelligible fashion and written in standard English?

Reviewer #1: Yes

Reviewer #2: Yes

Reviewer #3: Yes

5. Review Comments to the Author

Reviewer #1: Manuscript is well written and clearly describes the factors of land use changes in Japanese Society by comparing the factors affecting the farmland abandonment. Anyhow add more citations in support of your results in discussion section.

Reviewer #2: Comments

- The author need to provide a section on more recent literature

- In the discussion section, no comparison with the literature was developed

- The paper needs to elaborate more the contribution in terms of methodology and added value to the literature

Reviewer #3: This paper provides a detailed analysis of factors driving farmland abandonment in Hokkaido, Japan. Overall, i recommend that the discussion of the implications of the study be strengthened.

I recommend some minor modifications:

-the authors indicate that the study should be helpful to facilitate future land planning, but they say little about recommendations or policy implications of their findings and discussion. Do they have an overall assessment of the strengths and weaknesses of the polices through the various periods?

-as for ecological implications of farmland abandonment (and potential for recovery of ecosystems), this topic is mentioned a few times but there is no clear assessment or recommendations around this. The conclusion mentions two possible outcomes of reverting to natural ecosystems or becoming a wasteland. Could the authors comment on how common each of these is for the lands that have been abandoned to date?

-Reference 13 relates to "Rewilding abandoned landscapes in Europe". Has this been explicitly attempted in Japan?

-clarify why the analysis only extends to 2009

-how common is farmland abandonment in other parts of the world? What are the similarities and differences compared to the Japanese case/context?

6. PLOS authors have the option to publish the peer review history of their article (what does this mean?). If published, this will include your full peer review and any attached files.

Reviewer #1: No

Reviewer #2: No

Reviewer #3: Yes: Steffanie Scott

---

## [Author Response · Author response to Decision Letter 0]

31 May 2020

Dear Prof. Eda Ustaoglu,

Thank you for your letter concerning our manuscript. We received your E-mail on March 19th and have revised the manuscript. We were pleased to receive the favorable comments from the reviewers and have made revisions according to their comments.

1. We added the potential to extend our study to other studies in the Abstract.

(L21 – L22. Review’s comment 4) 

2. We have revised the first paragraph of the Introduction and removed Fig. 1 (Total population of the world, 1950-2010 (Estimates) and 2011-2100(Projections) provided by the United Nations (2014)).

 (L 30 – L33. Review’s comment 2).

3. We added an explanation on limiting the latest periods to include only the data available until 2009.

 (L146 – L152. Review’s comment 8).

4. We added the policy implications in the Discussion.

 (L393 – L401; L424 – L433. Reviewer’s comment 5)

5. We have revised the study implications in the Discussion.

(L434-437. Review’s comment 4)

6. We have revised the second paragraph of the Concluding remarks.

 (L445 – L461. Reviewer's comment 6).

7. We have revised the third paragraph of the Concluding remarks.

 (L462 – L475. Reviewer's comment 7).

8. We added some recent studies to compare previous studies investigating other countries and Japan.

(References 3, 6, 14, 16-21, 23, 71-91. Reviewer’s comment 1, 2, 3, 4, 9). 

9. We revised some words to clarify our description.

(highlighted in blue)

10. All the maps were created by our team based on data that have been made public. These public data have been referenced in our manuscript.

We would like to thank the reviewers for their helpful comments, and we hope that the revised manuscript is acceptable for publication in PLOS ONE.

Yours sincerely,

The Authors

Dear Reviewers

Thank you for constructive comments on our manuscript. We have revised the manuscript accordingly. 

Reviewer #1

1. Comment 1: Manuscript is well written and clearly describes the factors of land use changes in Japanese Society by comparing the factors affecting the farmland abandonment. Anyhow add more citations in support of your results in discussion section.

 We referenced additional studies to support our results in the discussion section. In particular, we have cited studies showing the important determinants of farmland abandonment in the population decline phase. These determinants include disadvantaged geographical conditions, depopulation, and low accessibility to markets.

Reviewer #2

2. Comment 2: The author need to provide a section on more recent literature

 We added recent literature in the Introduction, Discussion and Concluding remarks; thus, we revised the description of the world’s population dynamics (L30 – L33) and removed Fig. 1 (Total population of the world, 1950-2010 (Estimates) and 2011-2100 (Projections) provided by the United Nations (2014)) because a similar figure is found in the latest report of world population prospects provided by the United Nations (2019) (Reference 3).

3. Comment 3: In the discussion section, no comparison with the literature was developed

 We have referenced several studies and compared our findings with the results of previous studies in the Discussion (see the response to Comment 9 for details.).

4. Comment 4: The paper needs to elaborate more the contribution in terms of methodology and added value to the literature

 We have mentioned the potential and value of our results in the Abstract (L21 – L22) and Discussion (L434 – L437). Our results showing the land-use drivers associated with specific periods may provide new insights for other developed countries experiencing depopulation problems. We have also described the political implications of our results in the Concluding remarks (see the response to Comment 5 for details.). Since this is not a methodology paper, we did not discuss the methods.

Reviewer #3

5. Comment 5: the authors indicate that the study should be helpful to facilitate future land planning, but they say little about recommendations or policy implications of their findings and discussion. Do they have an overall assessment of the strengths and weaknesses of the polices through the various periods?

 We have discussed the political implications of our results (L393 – L401; L424 – L433). Our results showed that the disadvantaged farmlands, such as marginal, underproductive, narrow, and steep lands, will be continuously abandoned even in the decline phase. These geographical conditions were similar to eligible farmlands for the direct payment (DP) policy of subsidies to maintain unfavorable farmlands in place since 2000 in Japan. Thus, the current DP system might be effective even during the population decline phase. However, the population decline has become one of the determinants for farmland abandonment in recent years, and therefore, we expect that abandoned farmlands will increase in the future with depopulation. According to a previous study, population decentralization mitigates farmland abandonment even in the depopulation scenario; therefore, abandoned farmlands might be reduced using such a land management policy. The land management policy for population decentralization that was initiated in 2016 in Hokkaido would be a key policy for mitigating farmland abandonment.

6. Comment 6: as for ecological implications of farmland abandonment (and potential for recovery of ecosystems), this topic is mentioned a few times but there is no clear assessment or recommendations around this. The conclusion mentions two possible outcomes of reverting to natural ecosystems or becoming a wasteland. Could the authors comment on how common each of these is for the lands that have been abandoned to date?

 We have discussed the ecological implications in the Concluding remarks (L448 – L461). The previous studies showed that farmland abandonment can have either negative or positive ecological effects depending on the conservation target and regional circumstances. We have discussed previous studies to show that natural succession in abandoned farmland is considered an opportunity or a threat to regional biodiversity.

7. Comment 7: Reference 13 relates to "Rewilding abandoned landscapes in Europe". Has this been explicitly attempted in Japan?

 We have described attempts to rewild abandoned farmlands in Japan in the Concluding remarks (L462 – L475). In Japan, there have been very few attempts to rewild abandoned farmlands because vegetation recovery in abandoned farmlands is considered to be a threat to agrobiodiversity represented by traditional agricultural activities. However, there are several activities to restore abandoned farmlands to original natural ecosystems in areas with high potential and needs for rewilding.

8. Comment 8: clarify why the analysis only extends to 2009

 We have added an explanation for why we limited the latest period to include only data available up to 2009 in the Methods (L146 – L152). In Japan, the grid square system was used as a spatial unit for statistical use and/or biodiversity surveys beginning in 1969. However, there are two geographical summary grids based on old and new Japanese geodetic reference systems. Although the Japanese geodetic reference system was changed to the new one in 2002, the land-use data and population data had been continuously released using the two grids. However, the production of population data using old grid system was ended in 2010, and only data from the new grid system has been released since then. Therefore, we have to use the land-use datasets only up to 2009 to analyze the relationship between population and land-use using the same (old) grid system.

9. Comment 9: how common is farmland abandonment in other parts of the world? What are the similarities and differences compared to the Japanese case/context?

 We referenced studies regarding farmland abandonment in the Discussion (L391 – L392). Our results showed that geographical conditions, such as marginal, underproductive, narrow, and steep slopes, were determinants of an increase in abandoned farmlands during the growth and the decline phases, which was a similar trend found by the previous studies in other countries conducted in the growth phase. An original finding of the present study is that social conditions, such as the distance from densely inhabited districts (DIDs) and the population, exerted opposite effects between the population growth and decline phases.

---

## [Decision Letter · Decision Letter 1]

24 Jun 2020

Drivers of land-use changes in societies with decreasing populations: A comparison of the factors affecting farmland abandonment in a food production area in Japan

PONE-D-20-05452R1

Dear Dr. Kobayashi,

We’re pleased to inform you that your manuscript has been judged scientifically suitable for publication and will be formally accepted for publication once it meets all outstanding technical requirements.

Kind regards,

Eda Ustaoglu, PhD

Academic Editor

PLOS ONE

Additional Editor Comments (optional):

Reviewers' comments:

Reviewer's Responses to Questions

**Comments to the Author**

1. If the authors have adequately addressed your comments raised in a previous round of review and you feel that this manuscript is now acceptable for publication, you may indicate that here to bypass the “Comments to the Author” section, enter your conflict of interest statement in the “Confidential to Editor” section, and submit your "Accept" recommendation.

Reviewer #1: All comments have been addressed

Reviewer #2: All comments have been addressed

2. Is the manuscript technically sound, and do the data support the conclusions?

Reviewer #1: Yes

Reviewer #2: Yes

3. Has the statistical analysis been performed appropriately and rigorously? 

Reviewer #1: Yes

Reviewer #2: Yes

4. Have the authors made all data underlying the findings in their manuscript fully available?

Reviewer #1: Yes

Reviewer #2: (No Response)

5. Is the manuscript presented in an intelligible fashion and written in standard English?

Reviewer #1: Yes

Reviewer #2: Yes

6. Review Comments to the Author

Reviewer #1: Thank you for incorporating all the comments made by reviewers. Now the article is in intangible form and fulfill the creteria.

Reviewer #2: (No Response)

7. PLOS authors have the option to publish the peer review history of their article (what does this mean?). If published, this will include your full peer review and any attached files.

Reviewer #1: Yes: Umar Ijaz Ahmed

Reviewer #2: No

---

## [Editor Report · Acceptance letter]

30 Jun 2020

PONE-D-20-05452R1 

Drivers of land-use changes in societies with decreasing populations: A comparison of the factors affecting farmland abandonment in a food production area in Japan 

Dear Dr. Kobayashi:

I'm pleased to inform you that your manuscript has been deemed suitable for publication in PLOS ONE. Congratulations! Your manuscript is now with our production department. 

Kind regards, 

on behalf of

Dr. Eda Ustaoglu 

Academic Editor

PLOS ONE